# Species Delimitation and Genetic Relationship of *Castanopsis hainanensis* and *Castanopsis wenchangensis* (Fagaceae)

**DOI:** 10.3390/plants12203544

**Published:** 2023-10-12

**Authors:** Xing Chen, Yi Feng, Shuang Chen, Kai Yang, Xiangying Wen, Ye Sun

**Affiliations:** 1Guangdong Key Laboratory for Innovative Development and Utilization of Forest Plant Germplasm, College of Forestry and Landscape Architecture, South China Agricultural University, Guangzhou 510642, China; chenxinglf@163.com (X.C.); fm960104@163.com (Y.F.); chenshuang5512@163.com (S.C.); jerrykai1314@163.com (K.Y.); 2South China Botanical Garden, Chinese Academy of Sciences, Guangzhou 510650, China

**Keywords:** *Castanopsis hainanensis*, *Castanopsis wenchangensis*, microsatellite marker, chloroplast genome

## Abstract

*Castanopsis* is one of the most common genus of trees in subtropical evergreen broad-leaved forests and tropical monsoon rainforests in China. *Castanopsis hainanensis* and *Castanopsis wenchangensis* are endemic to Hainan Island, but they were once confused as the same species due to very similar morphologies. In this study, nuclear microsatellite markers and chloroplast genomes were used to delimit *C. hainanensis* and *C. wenchangensis*. The allelic variations of nuclear microsatellites revealed that *C. hainanensis* and *C. wenchangensis* were highly genetically differentiated with very limited gene admixture. Both showed higher genetic diversity within populations and lower genetic diversity among populations, and neither had further population genetic structure. Furthermore, *C. wenchangensis* and *C. hainanensis* had very different chloroplast genomes. The independent genetic units, very limited gene admixture, different distribution ranges, and distinct habitats all suggest that *C. wenchangensis* and *C. hainanensis* are independent species, thus they should be treated as distinct conservation units.

## 1. Introduction

Accurately identifying species identities and boundaries is necessary for determining conservation and management units [1]. There is much debate on defining species, and dozens of species concepts have been proposed, such as the morphological species concept that highlights intuitive differences in phenotypic characteristics, the biological species concept that emphasizes reproductive isolation, and the evolutionary species concept that considers independent evolutionary histories and ecological niches [2,3]. For trees, the difficulty in defining species is often exacerbated by factors such as longer generation times and larger effective population sizes, which may slow down lineage sorting [4]. In addition, frequent introgression may increase the chance of shared polymorphisms in markers or traits [5]. Although less contentious than taxonomic delimitation and the species-level debate, it is also complex and challenging to define conservation and management units for conservation actions [6,7].

*Castanopisis* species are dominant trees in subtropical evergreen broad-leaved forests and tropical monsoon rainforests [8]. The leaves of the *Castanopsis* tree are tough and sclerotized, with a thick cuticula, usually alternate and distichous, or rarely spirally arranged. The inflorescences are usually unisexual; male flowers are common in clusters, but female flowers are often solitary on erect catkins. The bracts are spinelike or rarely scalelike, sparsely or densely covered outside of cupules, and have diagnostic morphological characteristics in different species [9]. Although certain progress has been made in the taxonomy and inventory of *Castanopisis*, there are still many doubts about the classification and identification of some species [10].

*Castanopsis wenchangensis* G. A. Fu et Huang was first published in 1989 [11]; however, it is noted that the description of *C. wenchangensis* in FRPS (Flora Reipublicae Popularis Sinicae) was not the same as the original publication [10,11,12]. Due to this mistake, the identification of this species has been misled for many years. Because *C. wenchangensis* has highly similar morphologies to *C. hainanensis* Merr., which caused substantial confusion about the classification and identification of the two species, they were once considered the same species. Although the morphological characteristics of the two species are very similar, they can still be distinguished. *C. hainanensis* has relatively large cupules with a diameter of 4–5 cm, entirely covered with spines; the leaves are 5–12 cm long and 2.5–5 cm wide, rounded at the apex, acute at the base, and have 10–18 secondary veins on each side of the midvein [9]. In contrast, *C. wenchangensis* has relatively small cupules with a diameter of 1.4–2.0 cm, sparsely covered with gray pubescent spines; the leaves are 4.5–6.5 cm long and 1.9–3.1 cm wide, acuminate at the apex, obtuse at the base, and have 6–9 secondary veins on each side of the midvein [10,11]. It is worth mentioning that the leaves of *C. wenchangensis* are abaxially covered with dense pubescences and reddish-brown scales, which makes this species more prominent and recognizable in late autumn. Both *C. hainanensis* and *C. wenchangensis* are endemic species of Hainan Island; the former is distributed in the mountainous regions of the central and southern parts of Hainan Island, while the latter is only restricted to a very small area in the coastal lowlands of Wenchang City. Therefore, another question has been raised: Is *C. wenchangensis* an independent species or is it a distinct population of *C. hainanensis*? Thus, the investigation of the genetic differentiation between *C. hainanensis* and *C. wenchangensis* has critical importance for the full understanding of the delimitation and relationship between the two species.

At present, it has become very common to differentiate and delimit species by using nuclear and chloroplast variation. Simple sequence repeat (SSR) is one of the most commonly used molecular markers and has widespread application in the identification of population structure and conservation units [13,14]. Despite the highly conserved structures of the chloroplast genomes of plants, considerable sequence variations have been revealed within and between species, which has enhanced our understanding of plant diversity and evolutionary relationships [15]. The advances in high-throughput sequencing technologies and bioinformatic tools have facilitated rapid progress in the comparative analysis of chloroplast genomes and allowed species identification [16].

In this study, we used nuclear microsatellite loci and comparative chloroplast genomes to investigate the genetic differentiation between *C. hainanensis* and *C. wenchangensis*, with emphasis on delineating the species boundaries of *C. hainanensis* and *C. wenchangensis*, assessing the genetic diversity of existing populations, and providing effective conservation guidelines for the two endemic species.

## 2. Results

### 2.1. Nuclear SSR (nSSR) Diversity and Genetic Structure

Among the 16 nSSRs, four loci significantly deviated from the Hardy–Weinberg equilibrium (*p* < 0.01) and were excluded from further analysis. Genetic diversity was estimated for the 12 retained loci (Table 1). The number of alleles (A) per locus ranged from 3 to 11, with an average of 6.167. The expected heterozygosity (H_e_) was 0.169–0.742, and gene diversity in the total population was 0.176–0.823. The mean values of F_ST_, G_ST,_ and R_ST_ were 0.164, 0.149, and 0.157, respectively.

At the population level, the allele richness (A_R_) was 2.060–2.803 in *C. wenchangensis* and 2.833–3.331 in *C. hainanensis* (Table 2). Population SMX harbored the highest genetic diversity (A_R_ = 3.331), while PD had the lowest genetic diversity (A_R_ = 2.060). The expected heterozygosity (H_e_) ranged from 0.329 to 0.466 in *C. wenchangensis* and 0.433 to 0.476 in *C. hainanensis*. The mean heterozygosity of *C. wenchangensis* (H_e_ = 0.416) was lower than that of *C. hainanensis* (H_e_ = 0.454). The mean coefficient of inbreeding (F_IS_) was 0.007 and 0.05 in *C. wenchangensis* and *C. hainanensis*, respectively.

Genetic structure analysis of all samples revealed the optimal number of groups (K) was 2, indicating two genetic clusters that corresponded well to *C. wenchangensis* and *C. hainanensis* (Figure 1A). We made further genetic structure analyses for individuals of *C. wenchangensis* and *C. hainanensis* separately. There was no further population structure within both species; all individuals were equally admixed, as shown with K = 2, 3, and 4 (Figure 1B,C), suggesting each species was a single genetic cluster. The PCoA analysis showed the same results: the individuals of *C. wenchangensis* clearly separated from those of *C. hainanensis*, except for one sample in the LFT population (Figure 2). This individual should not be an introgressant since the two species have very limited gene admixture, as shown in Figure 1A. This is not a misnamed sample either, as the two species can be distinguished morphologically. The inconsistent results may be due to the different approach used, since PCoA is an ordination analysis based on similarity rather than a model-based clustering method.

AMOVA analysis of all samples showed that 24.3% of molecular variation existed between species, 2.67% occurred among populations within species, and 73.04% happened within populations (Table 3). In each species, molecular variation mainly existed within populations. The F_CT_ value was 0.24296, indicating a high degree of genetic differentiation between the two species.

### 2.2. Comparative Analysis of Chloroplast Genomes

Both chloroplast genomes of *C. hainanensis* and *C. wenchangensis* showed a typical quadripartite structure that consisted of the large single copy (LSC) region, the small single copy (SSC) region, and a pair of inverted repeat (IR) regions (IRa and IRb) (Figure 3 and Figure 4). The difference in total chloroplast genome length between *C. hainanensis* and *C. wenchangensis* was 229 bp; however, the genome structure and gene composition were highly similar between the two species. The total length of the chloroplast genomes of *C. hainanensis* and *C. wenchangensis* was 160,406 and 160,635 bp, respectively (Table 4). The chloroplast genome of each species had a total of 130 chloroplast genes, including 8 rRNA, 37 tRNA, and 85 protein-coding genes. The total GC content was 36.8% for both species, with the highest GC content of 42.8% in the IR region. The GC content of the LSC and SSC regions was slightly different, with 34.7% and 31% in *C. hainanensis*, and 34.6% and 30.09% in *C. wenchangensis*.

Genes on the four boundaries (LSC/IRb/SSC/IRa) were identical in *C. wenchangensis* and *C. hainanensis* (Figure 5). The ycf1 pseudogene straddled the JSB (IRb/SSC) boundary and showed different lengths in *C. wenchangensis* and *C. hainanensis*, with 1160 and 1151 bp, respectively. The ndhF gene was 2261 bp in *C. wenchangensis* and 2255 bp in *C. hainanensis*. The ycf1 gene spanned the JSA (SSC/IRa) boundary and had different lengths in *C. wenchangensis* and *C. hainanensis*, with 5684 and 5672 bp, respectively.

Four kinds of long repeats, including forward repeat, palindromic repeat, reverse repeat, and complement repeat, were detected in the chloroplast genomes of *C. hainanensis* and *C. wenchangensis*, but various types of repeats had different numbers in the two species. The chloroplast genome of *C. hainanensis* had 13 forward repeats, 11 palindromic repeats, 1 reverse repeat, and 1 complement repeat, while the chloroplast genome of *C. wenchangensis* possessed 22 forward repeats, 12 palindromic repeats, 3 reverse repeats, and 1 complement repeat.

## 3. Discussion

Nuclear SSR markers have been used to evaluate genetic differentiation among some species of the Fagaceae family, such as between *Quercus aquifolioides* and *Q. spinosa* (F_CT_ = 0.26) [17], among *Q. aliena*, *Q. dentata*, and *Q. variabilis* (F_CT_ = 0.21) [18], among *Q. acutissima*, *Q. variabilis,* and *Q. chenii* (F_CT_ = 0.195) [19], and between *Castanopsis sieboldii* and *C. cuspidate* (F_CT_ = 0.145) [20]. The genetic differences between different lineages within species have also been evaluated, but they are much lower than the genetic differentiation between species, as revealed by groups within *Q. aquifolioides* (F_CT_ = 0.04), *Q. spinose* (F_CT_ = 0.09), *C. sieboldii* (F_CT_ = 0.014), and *C. cuspidata* (F_CT_ = 0.095) [17,20]. In this study, we detect high and significant genetic differentiation between *C. wenchangensis* and *C. hainanensis* (F_CT_ = 0.243 in AMOVA analysis) based on allelic frequencies of nuclear SSRs, suggesting they are distinct species despite highly morphological similarities. Both the genetic structure and PCoA analyses reveal two distinct clusters, which correspond well to *C. wenchangensis* and *C. hainanensis*. The genetic structure analyses revealed only minimal admixture between the two gene pools, which may explain significant genetic differentiation between *C. wenchangensis* and *C. hainanensis* and imply their independent evolutionary histories. The demographically independent units characterized by restricted gene flow should be delimited as management units for biodiversity conservation [6,7,21]. So, distinguishing *C. wenchangensis* from *C. hainanensis* is of great significance for safeguarding genetic diversity. The distribution range of *C. hainanensis* and *C. wenchangensis* does not overlap. *C. hainanensis* is distributed in the central and southern parts of Hainan Island, while *C. wenchangensis* is restricted to a coastal area of northeastern Hainan Island. The habitat of *C. hainanensis* is mountainous red and loess soil, while *C. wenchangensis* occurs in coastal sandy soil. The independent genetic units, very limited genetic admixture, different distribution ranges, and distinct habitats further enhance the recognition of two independent species [5].

Consistent with this conclusion, we discover that the chloroplast genomes of *C. wenchangensis* and *C. hainanensis* are very different. Although the overall structure of the chloroplast genome is highly conserved, appreciable differences exist between *C. wenchangensis* and *C. hainanensis*. For example, different numbers of forward, palindromic, and reverse repeats are detected in the two chloroplast genomes, and the lengths of ndhF and ycf1 genes are not the same in the two species. The total length difference between the chloroplast genomes of *C. wenchangensis* and *C. hainanensis* is 229 bp. In a previous study, the difference in chloroplast genome length between Fagaceae species can reach 1336 bp, but the length difference between the two species in *Castanopsis* (*C. echidnocarpa* and *C. concinna*) was only 41 bp [22]. The considerable differences that exist in the chloroplast genomes between *C. wenchangensis* and *C. hainanensis* lead us to believe that they are two different species [23].

The higher the genetic diversity of a species, the greater its evolutionary potential and ability to adapt to the environment. Such species have a higher probability of long-term survival when environmental conditions change [24]. In this study, the nuclear SSR genetic diversity revealed in *C. hainanensis* (A = 3.467, A_R_ = 3.062, H_O_ = 0.450, H_e_ = 0.454) is slightly higher than that in *C. wenchangensis* (A = 3.042, A_R_ = 2.626, H_O_ = 0.424, H_e_ = 0.416), suggesting that the two species have similar potential for adapting to environmental changes. Compared with other species in the genus *Castanopsis* such as *C. acuminatissima* (A = 10.8, H_e_ = 0.716) [25] and *C. sieboldii* (A = 5.226, H_e_ = 0.574) [26], the genetic diversity levels of *C. hainanensis* and *C. wenchangensis* are relatively low, which may be due to the small geographic range of the two species [27]. This is consistent with the previous results that rare plant species often show a lower level of genetic diversity than widespread congeners [28,29]. This also means that the conservation of the two species needs to be strengthened. Both *C. hainanensis* and *C. wenchangensis* are endemic to Hainan Island, but they have different habitats, indicating that they have precious gene resources and underlying ecological adaptations that are worth investigating more in depth. It is interesting to find that *C. hainanensis* exhibits a high degree of genetic uniformity rather than highly structured isolated populations, although it has a wider distribution than *C. wenchangensis*. AMOVA analysis reveals only negligible genetic differences among populations within each species (F_ST_ = 0.04 in *C. hainanensis* and F_ST_ = 0.05 in *C. wenchangensis*). Furthermore, in terms of how genetic variation is partitioned within and among populations, there does not appear to be a difference between *C. hainanensis* and *C. wenchangensis*; both show higher genetic diversity within populations and lower genetic diversity among populations, which is in accordance with the general patterns of genetic diversity in tree species [27,30] and strengthens our view that *C. hainanensis* and *C. wenchangensis* are independent species.

## 4. Materials and Methods

### 4.1. Sampling and DNA Extraction

A total of 170 individuals from 11 natural populations of *C. hainanensis* and *C. wenchangensis* were collected on Hainan Island (Figure 6; Table 5). Sampling individuals were kept at least 20 m apart to avoid close relatives. Fresh leaves were collected and immediately dried with silica gel. Genomic DNA was extracted from the silica gel-dried leaves using the Tiangen Plant Genomic DNA Extraction Kit (DP320) according to the instructions of the manufacturer. DNA quality was determined by 1.00% (*w*/*v*) agarose gel.

### 4.2. Nuclear SSR Genotyping and Genetic Structure Analysis

A total of 16 primer pairs of nuclear SSRs (Table 6) were screened from those reported in *Castanopsis* and Castanea species [31,32,33]. Quadruple fluorescent polymerase chain reaction (PCR) was amplified using the Type-it microsatellite PCR kit (QIAGEN, Hilden, Germany). PCR was performed in a mixture including 20 ng of genomic DNA, 1× PCR Master Mix, 1× Q-Solution, and 10 μM of each primer (forward and reverse). The forward primers were labeled with different fluorescent dyes. The PCR program was set as follows: 95 °C for 5 min, followed by 28 cycles of 95 °C for 30 s, 57 °C for 90 s, and 72 °C for 30 s, and a final extension at 60 °C for 30 min. The PCR products were separated by capillary electrophoresis with the ABI-3730XL fluorescence sequencer (Applied Biosystems, Foster City, CA, USA), using LIZ500 as the internal standard. Alleles were scored using Genemarker2.2.0 [34].

FSTAT 2.9.4 [35] and GeneALEx 6.5 [36] were used to calculate the number of alleles (A) observed, allelic richness (A_R_), observed heterozygosity (H_O_), expected heterozygosity (H_e_), genetic diversity within populations (H_S_), total genetic diversity (H_T_), inbreeding coefficient (F_IS_), and genetic differentiation among populations (G_ST_, F_ST_, and R_ST_). Deviations from the Hardy–Weinberg equilibrium (HWE) were tested with 1000 permutations, and nSSRs deviating from HWE were excluded from further analysis.

Population genetic structure analysis was performed using STRUCTURE 2.3.4 [37]. The length of the burn-in period was set to 1,000,000, and MCMC replications after the burn-in were set to 500,000. Numbers of assumed clusters (K) were assessed by performing 10 independent runs for each K value from 1 to 11. The optimal K value was selected according to the Evanno method using STRUCTURE HARVESTER [38]. The 10 runs of structure analysis were averaged using CLUMPP1.1.2 [39], and the results were graphically represented using Distruct 1.1 [40]. Principal coordinate analysis (PCoA) was performed to detect genetic distance among populations using GeneALEx 6.5 [36]. Analysis of molecular variance (AMOVA) was performed using ARLEQUIN v3.5 [41] to determine the proportion of genetic variation partitioned at the species, population, and intrapopulation levels.

### 4.3. Comparative Analysis of Chloroplast Genomes

Whole genome resequencing was performed for one sample of *C. wenchangensis* and *C. hainanensis* each. DNA library construction and Illumina paired-end sequencing were conducted by Nextomics Biosciences (Wuhan). The chloroplast genome was assembled using the software GetOrganelle 1.7.5.3 [16], and the parameters were adjusted appropriately to obtain the complete cyclic structure. The chloroplast genome was imported into the NCBI database for comparison and selection of the most suitable reference genome. Protein-coding genes, tRNAs, and rRNAs were annotated using the online software CPGAVAS2 [42]. Geneious v9.0.2 [43] was used to manually correct and complement problematic annotations. Finally, the resulting annotation files were used to generate a physical map of the chloroplast genome using OGDRAW [44].

The junction sites of the chloroplast genome were visualized with the online program IRscope (https://ir-scope.shinyapps.io/irapp/ (accessed on 17 October 2022)) [45]. Long repeats, including forward, reverse, palindrome, and complementary repeats, in chloroplast genomes were identified using REPuter (http://bibiserv.techfak.uni-bielefeld.de/reputer/ (accessed on 29 October 2022)) [46]. The parameters were set at a minimal repeat size of 30 bp and a Hamming distance of 3.

## 5. Conclusions

*C. hainanensis* and *C. wenchangensis* are endemic to Hainan Island; the former is distributed in the mountainous regions of the central and southern parts of Hainan Island, while the latter is restricted to the coastal lowlands of Wenchang City. They were once confused as the same species due to their very similar morphologies. Furthermore, *C. wenchangensis* has been wrongly identified because its description in FRPS is not the same as the original publication. Correctly identifying *C. hainanensis* and *C. wenchangensis* and determining their genetic diversity and genetic relationships is of great significance for the conservation and sustainable utilization of these two species.

In this study, we use nuclear microsatellite markers and comparative chloroplast genomes to determine the delimitation between *C. hainanensis* and *C. wenchangensis*. Based on the nuclear microsatellite variations, we demonstrate that *C. hainanensis* and *C. wenchangensis* are distinct clusters that are highly genetically differentiated with very limited genetic admixture; both showed higher genetic diversity within populations and lower genetic diversity among populations, and neither has further population genetic structure. Furthermore, this work allows us to distinguish the two species based on their chloroplast genomes. The independent genetic units, extremely limited genetic admixture, different distribution ranges, and distinct habitats make us believe that *C. wenchangensis* and *C. hainanensis* are independent species, thus they should be treated as different conservation units.

## Figures and Tables

**Figure 1 plants-12-03544-f001:**
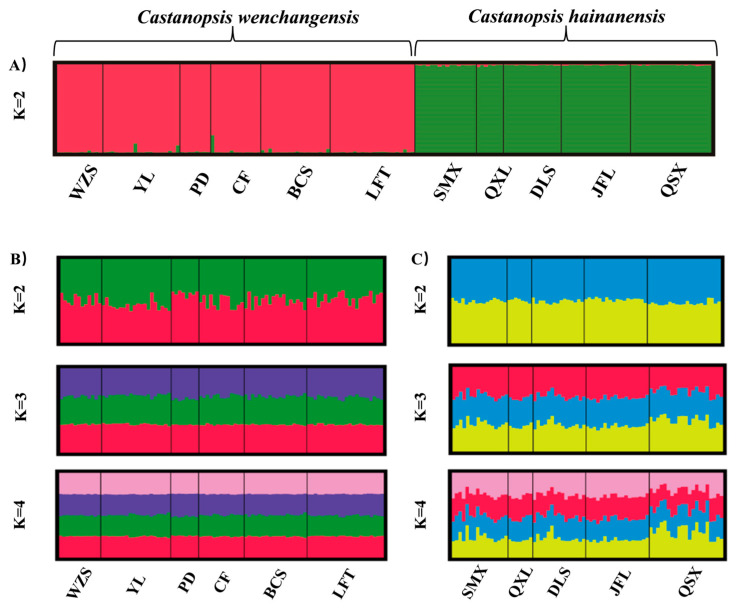
Bayesian clustering plot for all samples based on 12 nSSRs (**A**). Individual proportions of the membership in *C. wenchangensis* are shown when two, three, or four clusters were defined by genetic structure analyses, respectively (**B**). Individual proportions of the membership in *C. hainanensis* are shown when two, three, or four clusters were defined by genetic structure analyses, respectively (**C**).

**Figure 2 plants-12-03544-f002:**
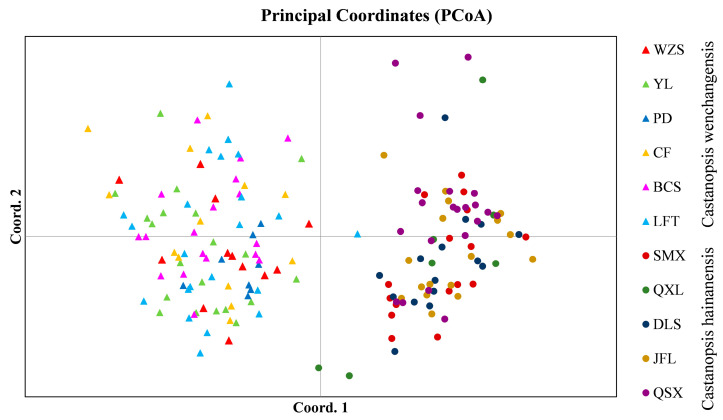
PCoA analysis for all samples based on 12 nSSRs.

**Figure 3 plants-12-03544-f003:**
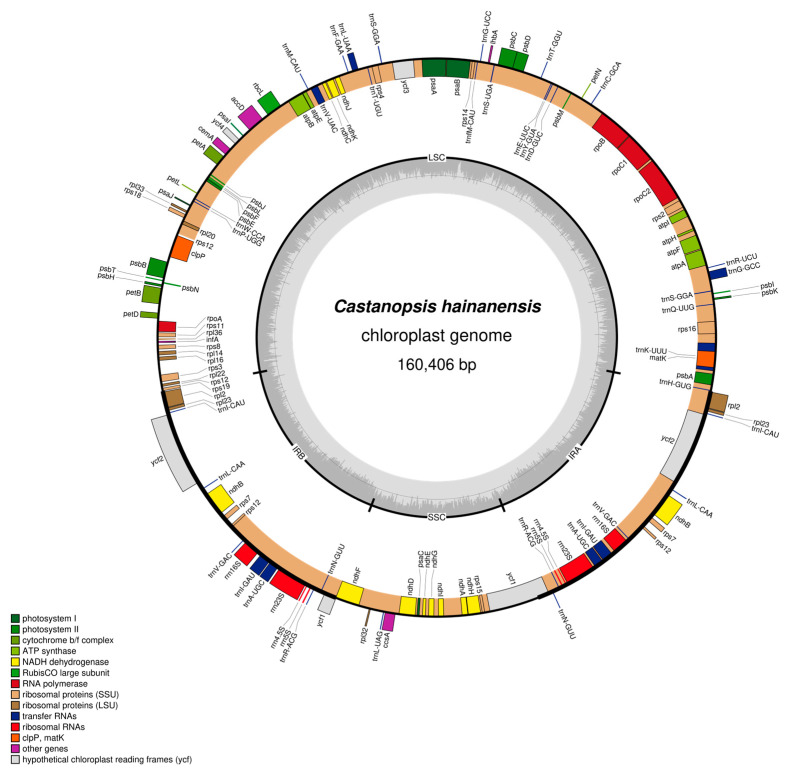
Physical map of the chloroplast genome of *C. hainanensis*. Genes inside the circles are transcribed clockwise, and those outside the circles are transcribed counterclockwise. Genes belonging to different functional groups are color-coded. The darker gray in the inner circle corresponds to GC content, while the lighter gray corresponds to AT content.

**Figure 4 plants-12-03544-f004:**
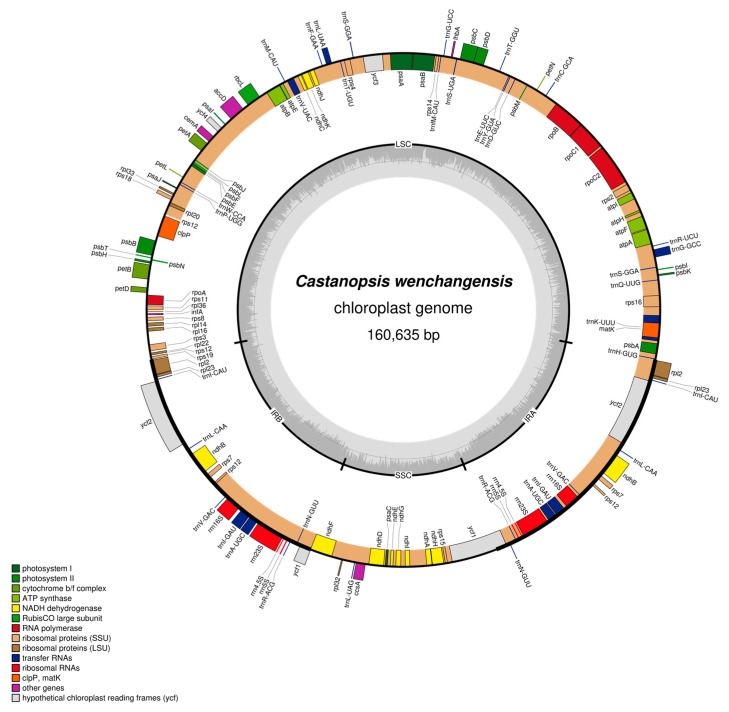
Physical map of the chloroplast genome of *C. wenchangensis*. Genes inside the circles are transcribed clockwise, and those outside the circles are transcribed counterclockwise. Genes belonging to different functional groups are color-coded. The darker gray in the inner circle corresponds to GC content, while the lighter gray corresponds to AT content.

**Figure 5 plants-12-03544-f005:**
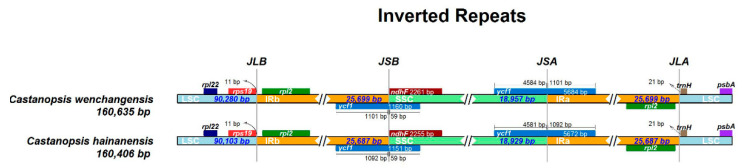
The four boundaries of the chloroplast genomes of *C. hainanensis* and *C. wenchangensis*.

**Figure 6 plants-12-03544-f006:**
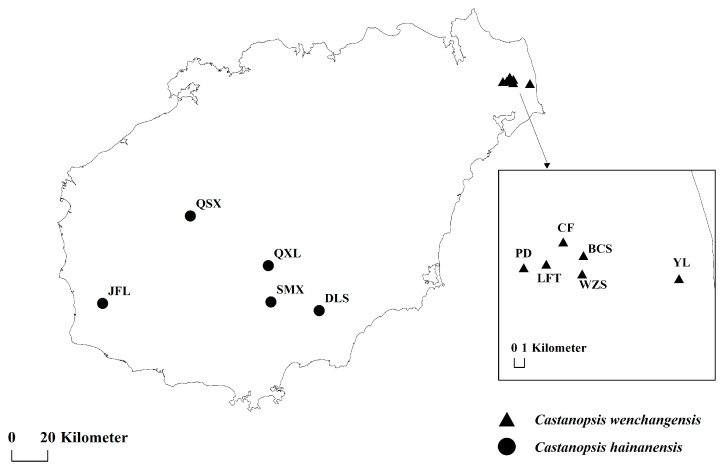
Sampling locations of *C. wenchangensis* and *C. hainanensis*.

**Table 1 plants-12-03544-t001:** Genetic diversity parameters at the 12 retained microsatellite loci.

Locus	A	H_O_	H_e_	H_S_	H_T_	F_IS_	F_ST_	G_ST_	R_ST_
CS92	11	0.697	0.700	0.728	0.82	0.024	0.103	0.112	0.143
CC-20303	5	0.24	0.279	0.292	0.328	0.169	0.135	0.109	0.028
CS24	7	0.546	0.503	0.521	0.6	−0.028	0.142	0.131	0.122
CC-30080	3	0.351	0.399	0.416	0.508	0.163	0.228	0.181	0.173
CC-935	5	0.414	0.401	0.417	0.438	0.01	0.045	0.049	0.129
CS20	10	0.719	0.742	0.772	0.823	0.054	0.06	0.063	0.143
CC-39198	4	0.304	0.336	0.351	0.364	0.114	0.048	0.036	0.089
CC4323	5	0.187	0.169	0.175	0.176	−0.063	0.01	0.009	0.025
CC-7378	6	0.448	0.393	0.407	0.569	−0.055	0.325	0.285	0.257
CC-11089	6	0.233	0.235	0.244	0.619	−0.015	0.63	0.606	0.496
CC-43042	7	0.492	0.527	0.55	0.634	0.124	0.151	0.133	0.122
CC-704	5	0.6	0.516	0.534	0.579	−0.1	0.089	0.078	0.162
Mean	6.167	0.436	0.433	0.451	0.538	0.033	0.164	0.149	0.157

A: alleles observed; H_O_: observed heterozygosity; H_e_: expected heterozygosity; H_S_: gene diversity within populations; H_T_: gene diversity in the total population; F_IS_: inbreeding index; F_ST_: genetic differentiation among populations; G_ST_: the proportion of total genetic diversity that occurred among the population; R_ST_: genetic differentiation among populations under a stepwise mutation model.

**Table 2 plants-12-03544-t002:** Genetic diversity parameters in 11 populations of the two species.

Species	Population	A	A_R_	H_O_	H_e_	F_IS_
*C. wenchangensis*	WZS	2.917	2.622	0.438	0.433	0.032
	YL	3.500	2.791	0.329	0.386	0.173
	PD	2.083	2.060	0.438	0.329	−0.267
	CF	3.083	2.799	0.429	0.466	0.119
	BCS	3.417	2.803	0.477	0.447	−0.039
	LFT	3.250	2.681	0.436	0.435	0.023
	Mean	3.042	2.626	0.424	0.416	0.007
*C. hainanensis*	SMX	3.917	3.331	0.510	0.469	−0.057
	QXL	2.833	2.833	0.405	0.450	0.176
	DLS	3.500	3.011	0.428	0.433	0.047
	JFL	3.500	3.023	0.417	0.443	0.088
	QSX	3.583	3.111	0.488	0.476	−0.002
	Mean	3.467	3.062	0.450	0.454	0.050

A: average number of alleles; A_R_: allele richness; H_O_: observed heterozygosity; H_e_: expected heterozygosity; F_IS_: inbreeding index.

**Table 3 plants-12-03544-t003:** AMOVA analyses based on 12 nSSRs.

Sample	Source of Variation	Sum of Squares	Variance Components	Percentage of Variation	Fixation Indices (*p* < 0.001)
*C. hainanensis* and *C. wenchangensis*	Among species	158.053	0.90204	24.3	F_CT_ = 0.24296 F_SC_ = 0.03525 F_ST_ = 0.26965
	Among populations within species	51.378	0.09907	2.67	
	Within populations	892.099	2.71155	73.04	
*C. hainanensis*	Among populations	25.778	0.1202	4.08	F_ST_ = 0.04078
	Within populations	421.306	2.82756	95.92	
*C. wenchangensis*	Among populations	25.600	0.08243	3.06	F_ST_ = 0.03055
	Within populations	470.793	2.61552	96.94	

**Table 4 plants-12-03544-t004:** Characteristics of the chloroplast genomes of *C. hainanensis* and *C. wenchangensis*.

Species	Total		LSC		SSC		IR		Gene			
	Length (bp)	GC (%)	Length (bp)	GC (%)	Length (bp)	GC (%)	Length (bp)	GC (%)	rRNA	tRNA	CDS	Total
*C. hainanensis*	160,406	36.8	90,103	34.7	18,929	31	25,687	42.8	8	37	85	130
*C. wenchangensis*	160,635	36.8	90,280	34.6	18,957	30.9	25,699	42.8	8	37	85	130

**Table 5 plants-12-03544-t005:** Sampling locations and sizes of 11 populations of *C. wenchangensis* and *C. hainanensis*.

Species	Sampling Location (Population Code)	Number of Individuals	Altitude (m)	Longitude (E)	Latitude (N)
*C. wenchangensis*	Wenzaoshan Village (WZS)	12	36	110°50′52.66″	19°47′33.27″
	Po Dui Village (PD)	8	28	110°49′49.78″	19°47′52.74″
	Longfei Tou Village (LFT)	22	27	110°51′0.07″	19°48′3.49″
	Ya Lang Village (YL)	20	38	110°57′56.49″	19°47′18.30″
	Chang Fa Village (CF)	13	40	110°51′53.05″	19°49′13.30″
	Baocaishan Village (BCS)	18	38	110°52′56.88″	19°48′30.47″
*C. hainanensis*	Shuiman Village (SMX)	16	658	109°40′33.11″	18°42′12.08″
	Qixianling Mountain (QXL)	7	264	109°39′46.51″	18°52′58.66″
	Diaoluo Mountain (DLS)	15	377	109°54′57.49″	18°39′35.21″
	Qingsong Village (QSX)	21	396	109°16′28.23″	19°7′49.38″
	Jianfengling Mountain (JFL)	18	283	108°50′16.43″	18°41′44.58″

**Table 6 plants-12-03544-t006:** The 16 nuclear SSR primer pairs used in this study.

Locus	Repetitive Unit	Primer Sequence (5′→3′)	Fragment Length
CC22256	(ACA)_9_	F: <TAMRA> CAAGTCCGATCCTTCCTCTG	93–111
		R: AGCTGGGTTTTGAGTAGCGA	
CS92	(GA)_12_…(AT)_3_	F: <HEX> CAGAAACCAAAAAAGAACAG	140–162
		R: ACACACAAGAAAACAAAAGC	
CC25435	(ACA)_11_	F: <6-FAM> TGAAAATCCTCTGGGTCTGG	250–268
		R: CTTCTCGCACAACATCCTCA	
CC20303	(TGT)_6_	F: <ROX> AGTGGTGGTGTTTCCCAAAG	204–225
		R: AGAAGAGCTTCCTTCCCCTG	
CS24	(CAA)_6_	F: <TAMRA> ATCACCGGAGAAAACCCTAACGA	121–142
		R: AATGTTTCGGACCAATTCGAGGT	
CC30080	(TTG)_4_	F: <HEX> CTCAGATCCGACCGTTTGTT	158–167
		R: ATGGGAGGATGGAAGGTAGG	
CC935	(TC)_6_	F: <TAMRA> TGCTGAGTTTCTGAGGCTGA	124–138
		R: GACACGTCGAATGGGAATCT	
CS20	(AG)_13_	F: <ROX> AATTTCACATCCCAACTCTGCGA	252–280
		R: TGGAGGGAGTAGTGGACGATCAA	
CC39198	(AG)_11_	F: <6-FAM> GGTTGTTGTCGTTGTCGTTG	204–232
		R: TCTGTCTCCGTTCACCCTCT	
CC4323	(TGT)_7_	F: <6-FAM> TCGGTACAACTTCTGGGTCC	238–253
		R: AGCCTCTTCTCCACAACGAA	
CC7378	(CCG)_5_	F: <HEX> CACTCTCTCCGGTCCATGAT	149–170
		R: AATGTGGCGAGTTCGGTAAC	
CC34976	(GA)_7_	F: <ROX> GTGGTGGATTTTGGGTATGG	260–290
		R: TCCCAAACCTTGTCACCTTC	
CC11089	(AGA)_12_	F: <TAMRA> CAGAACCAGTTTCGTGCTCA	115–139
		R: GCTTCTTGGTGGTGCTCTTC	
CC-43042	(CGC)_4_	F: <HEX> TAACCAATCACGTTCACCGA	193–211
		R: CGCCACATCTAAAACCCCTA	
CC-41684	(ACC)_6_	F: <ROX> ATCCTCCAAGCAATCCTCCT	279–306
		R: TCAAGTGTGTGCGAGTGACA	
CC-704	(GTT)_4_	F: <6-FAM> ATGCCTTGCTTCTCAGCATT	261–279
		R: CCAACAATAATGCCCCATTC	

## Data Availability

The chloroplast genome sequences are deposited in the GenBank, accession numbers: OR543091, OR545365.

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
