# Peer review of "Species Delimitation and Genetic Relationship of Castanopsis hainanensis and Castanopsis wenchangensis (Fagaceae)"

_plants, 2023, doi:10.3390/plants12203544_

Round 1

Reviewer 1 Report

Dear authors:

I have some comments about your interesting manuscript. Yours faithfully.

“The PCoA 114 analysis showed the same results that the individuals of C. wenchangensis clearly sepa-115 rated from that of C. hainanensis except for one sample in population LFT (Fig. 2).” Could you explain what was this sample? an introgressant or a misnamed sample? Did you reanalise the data without the odd sample?

Could you please include a general description of both species with illustrations? It is not easy to find botanical descriptions of them; even more, you cited some studies showing different morphological aspects for the same species. It would be great if you could show some pictures of each species checked in your genetic study.

Please, improve the discussion comparing the Fst obtained in this study with other studies about the genetic differentiation of species; had they similar Fct than in this study? “In this study, we clarify significant genetic differentiation (FCT=0.24296 in AMOVA 178 analysis) between C. wenchangensis and C. hainanensis by using allelic frequencies of nu-179 clear SSRs, suggesting they are distinct groups despite morphological similarities.”

Could you please improve the discussion on this paragraph? “Consistent with this conclusion, we discover that chloroplast genomes of the two 195 species are not same, with 229 bp of difference in total length. Although the overall 196 structure of chloroplast genomes is highly conserved, appreciable differences exist in the 197 two species, for example, different numbers of forward, palindromic, and reverse repeats 198 are detected in chloroplast genomes of the two species. Additionally, some genes such as 199 ndhF and ycf1 are not in same length in the two species. Compared with other subtrop0 ical and tropical trees [17,18], there are sufficient differences of chloroplast genomes be-201 Plants 2023, 12, x FOR PEER REVIEW 9 of 15 tween C. wenchangensis and C. hainanensis, leading us to believe that they are two differ-202 ent species.” When I checked on the reference 17 (Zhang, Y.T.; Huang, J.; Song, J.; Lin, L.M.; Feng, R.X.; Xing, Z.B. Structure and Variation Analysis of Chloroplast Genomes in 341 Fagaceae. Bulletin of Botanical Research 2018, 38, 757–765. ), did they showed higher differences among species?: “The chloroplast genomes of 14 Fagaceae plants were double-stranded circular structure with a size of about 160 kB with a small difference, with a maximum difference of only 1 366 bp. The order of the genes was basically the same, but the number of genes was different. infApetGrpl22ycf1, ycf15 and many other genes were lost in some species.”

Don’t you have any conclusion? You have it in the abstract but not in the main text.

Reviewer 2 Report

1. What is the main question addressed by the research?
The authors used nuclear microsatellite loci and comparative chloroplast genomes to investigate the genetic differentiation between C. hainanensis and C. wenchangensis.
2. Do you consider the topic original or relevant in the field? Does it address a specific gap in the field?
3. What does it add to the subject area compared with other published material?
Castanopsis hainanensis and Castanopsis wenchangensis are endemic on Hainan Island, but they are once confused as the same species due to very similar morphologies. This work allows us to distinguish the two species based on their chloroplast genome.
4. What specific improvements should the authors consider regarding the methodology? What further controls should be considered?
The methods section is well written.
5. Are the conclusions consistent with the evidence and arguments presented and do they address the main question posed?
The conclusion could be expanded a bit, but overall it is well written.
6. Are the references appropriate?
The references are appropriate.
7. Please include any additional comments on the tables and figures.
Overall, the article is well written and illustrated with figures and tables. Figure 1 needs improvement in terms of overall design and image quality.

The Quality of English Language is good. The article is written in good academic language.
